# MASSIVE EDITING FOR LARGE LANGUAGE MODELS VIA META LEARNING

**Chenmien Tan**[1]***, Ge Zhang**[23]*, **Jie Fu**[4]†

University of Edinburgh[1], University of Waterloo[2], 01.AI[3], HKUST[4]
`chenmien.tan@ed.ac.uk, gezhang@umich.edu, jiefu@ust.hk`

## ABSTRACT

While large language models (LLMs) have enabled learning knowledge from the pre-training corpora, the acquired knowledge may be fundamentally incorrect or outdated over time, which necessitates rectifying the knowledge of the language model (LM) after the training. A promising approach involves employing a hyper-network to generate parameter shift, whereas existing hyper-networks suffer from inferior scalability in synchronous editing operation amount (Hase et al., 2023b; Huang et al., 2023). For instance, Mitchell et al. (2022) mimic gradient accumulation to sum the parameter shifts together, which lacks statistical significance and is prone to cancellation effect. To mitigate the problem, we propose the **MA**ssive **L**anguage **M**odel **E**diting **N**etwork (MALMEN), which formulates the parameter shift aggregation as the least square problem, subsequently updating the LM parameters using the normal equation. To accommodate editing multiple facts simultaneously with limited memory budgets, we separate the computation on the hyper-network and LM, enabling arbitrary batch size on both neural networks. Our method is evaluated by editing up to thousands of facts on LMs with different architectures, *i.e.*, BERT-base, GPT-2, T5-XL (2.8B), and GPT-J (6B), across various knowledge-intensive NLP tasks, *i.e.*, closed book fact-checking and question answering. Remarkably, MALMEN is capable of editing hundreds of times more facts than MEND (Mitchell et al., 2022) with the identical hyper-network architecture and outperforms editor specifically designed for GPT, *i.e.*, MEMIT (Meng et al., 2023). Our code is available at `https://github.com/ChenmienTan/malmen`.

## 1 INTRODUCTION

Large language models (LLMs) have exhibited the ability to acquire knowledge from pre-training corpora and demonstrated promising performance in knowledge-intensive NLP tasks such as fact verification and question answering (Thorne et al., 2018; Petroni et al., 2019; Roberts et al., 2020). However, such knowledge may be factually incorrect and outdated over time. For instance, a language model (LM) trained before 2023 probably predicts "Paris Saint-Germain" rather than "Inter Miami CF" when prompted with "What sports team does Messi play for?". A straightforward remedy is to fine-tune the model on the corrected datasets, whereas such an approach suffers the risk of overfitting and catastrophic forgetting (Kirkpatrick et al., 2017; Zhu et al., 2020). It is challenging to edit the model precisely as the knowledge is implicitly and distributionally encoded in the parameters of LM. An ideal editing method is expected to be (i) *generalizable*, where the model behaves consistently across a different formulation of an injected fact and (ii) *local*, where the updates do not affect the remainder of the acquired knowledge (De Cao et al., 2021; Mitchell et al., 2022).

Several lines of research have investigated editing LM effectively and locally (Dai et al., 2022; Huang et al., 2023), among which an emerging approach entails training a hyper-network to generate updated parameters (Ha et al., 2017; De Cao et al., 2021; Hase et al., 2023b; Mitchell et al., 2022). Unlike fine-tuning, hyper-networks can explicitly designate editing generalizability and locality as objectives, learning to uphold the consistency and reliability of the LM (De Cao et al., 2021; Hase

---

*Work done while interning at HKUST.
†Corresponding author.

et al., 2023b; Mitchell et al., 2022). Recognizing that pre-trained weight is a good initialization, De Cao et al. (2021); Hase et al. (2023b); Mitchell et al. (2022) predict the parameter shift rather than directly generating the updated parameter. Existing hyper-networks also generate the parameter shift conditioned on the standard fine-tuning gradient as the gradient serves as a viable starting point for model editing and provides rich insights into how knowledge is encoded within the LM (De Cao et al., 2021; Hase et al., 2023b; Mitchell et al., 2022).

Although existing hyper-networks have shown notable performance in editing a single or few facts, they exhibit limited scalability in synchronous editing operation amount (Hase et al., 2023b; Huang et al., 2023). Due to the consideration of computation complexity, the language model is typically frozen when training the hyper-networks (De Cao et al., 2021; Hase et al., 2023b; Mitchell et al., 2022). A following downside is that the hyper-networks tend to overfit to the present state of the LM, allowing for only a few updates before necessitating re-training to adapt to the LM's new state, which renders meta-learning computationally expensive for mass editing in actual industrial scenarios (Hase et al., 2023b; Huang et al., 2023). Hase et al. (2023b) sought to extend meta-learning to sequential editing but only scale to few, *e.g.*, 10, updates. In this paper, we focus on another direction, which is to edit multiple facts at once so that the cost of training the hyper-network amortized to each editing is lower.

There are mainly two challenges to the issue: (i) the impact of varying edits on the model parameters may be contradictory, making it difficult to ascertain the parameters effective for all facts to be injected (Yu et al., 2020); (ii) training hyper-networks to edit a large number of facts simultaneously results in substantial memory consumption, potentially exceeding the hardware limit. Regarding the first challenge, existing hyper-networks resort to mimicking gradient accumulation to sum the parameter shifts together, which lacks statistical significance and is prone to cancellation effect (Mitchell et al., 2022). In contrast, we formulate the parameter shift aggregation as a least square problem to seek for the parameter shift effective for all facts to be injected. For the second challenge, instead of concatenating the hyper-network to the LM, we delineate the computation between the hyper-network and LM. The decomposition permits arbitrary batch sizes on both neural networks, significantly reducing the memory required.

The primary contribution of this work is an LM editing algorithm called MAssive Language Model Editing Network (MALMEN), which is designed for scalability across numerous facts while maintaining commendable editing performance and economical memory consumption. Empirical evaluations are conducted to edit up to thousands of facts on LMs with diverse architectures, *i.e.*, BERT-base (Devlin et al., 2019), GPT-2 (Radford et al., 2019), T5-XL (2.8B; Raffel et al., 2020), and GPT-J (6B; Wang & Komatsuzaki, 2021) across various knowledge-intensive NLP tasks, *i.e.*, closed book fact verification and question answering. We also perform ablation studies to elucidate the effect of design components of MALMEN.

## 2 RELATED WORK

Several lines of research have investigated on model editing, including fine-tuning with hard constraint (Zhu et al., 2020), editing with external memory (Mitchell et al., 2023; Hartvigsen et al., 2023; Huang et al., 2023), locate-and-edit (Dai et al., 2022; Gupta et al., 2023; Hase et al., 2023a), and meta-learning (Sinitsin et al., 2020; De Cao et al., 2021; Mitchell et al., 2022). We refer readers to Yao et al. (2023) for a comprehensive survey.

**Editing with External Memory**   Mitchell et al. (2023) store edits in an explicit memory, utilizing a scope classifier to evaluate the probability that an input falling within the realm of a stored edit point. If the input matches any point in the storage, the counter-factual model generates output conditioned on the input and edit point. Huang et al. (2023) address an LM error by incorporating an additional neuron into the feed-forward network (FFN) of the last Transformer block. By leveraging the sparsity of GeLU activation function (Hendrycks & Gimpel, 2016), the neuron is trained to be only valid for in-scope inputs, then the inserted neuron does not alter the output of unrelated input.

**Locate-and-edit**   Dai et al. (2022) employ integrated gradients (Sundararajan et al., 2017) to pinpoint the location of knowledge within LM at the neuron level. Drawing inspiration from Geva et al. (2021), Dai et al. (2022) hand-craft modifications to the value slots corresponding to knowledge

neurons to rectify LM outputs. Meng et al. (2022; 2023) measure the efficacy of hidden states in GPT for fact recall through causal tracing (Vig et al., 2020), where the representation of the subject's last token within the FFN at intermediate layers to be significant. On the basis, Meng et al. (2022; 2023) conceptualize the linear layers as a key-value memory association and modify the value of effective hidden states. However, Hase et al. (2023a) observe that representation denoising provides limited insights into the best layer for model editing.

**Meta-learning** Sinitsin et al. (2020) describe a bi-level meta-learning that identifies model initialization for quick fine-tuning. De Cao et al. (2021); Mitchell et al. (2022) learn to transform the standard fine-tuning gradient to a more targeted parameter shift, where they mainly focus on building light-weight hyper-networks as a naive multi-layer perception (MLP) that intakes the gradients and outputs the parameter shifts of a linear layer suffer a quadratic complexity with respect to the hidden size. Specifically, De Cao et al. (2021) pose LSTM (Hochreiter & Schmidhuber, 1997) to project the sentence embedding into rank-1 masks (Krueger et al., 2017) over the gradients; Mitchell et al. (2022) decompose the gradient into the outer product of keys and value gradients and apply low-rank MLPs (Hu et al., 2022) to refine the keys and value gradients.

## 3 PROBLEM FORMULATION

Let $\mathcal{X}$ be the prompt set and $\mathcal{Y}$ be the answer set, *e.g.*, for fact verification, the answer set is binary, *i.e.*, {True, False}; for question answering, the answer set is the vocabulary set. For each *edit* prompt-answer tuple $(x, y) \in \mathcal{X} \times \mathcal{Y}$, let

- $E(x, y) \subseteq \mathcal{X} \times \mathcal{Y}$ be the *equivalent* tuple collection subject to $(x, y)$ such that $(x^e, y^e)$ is semantically equivalent to $(x, y)$ for any $(x^e, y^e) \in E(x, y)$. For example, for the edit tuple ("What is the capital of China?", "Beijing."), an equivalent tuple is ("Of what country is Beijing the capital?", "China.");
- $U(x, y) = (\mathcal{X} \times \mathcal{Y}) \backslash E(x, y)$ be the *unrelated* tuple collection subject to $(x, y)$, where $(x^u, y^u)$ is unrelated to $(x, y)$ for any $(x^u, y^u) \in U(x, y)$. For the edit tuple example above, an unrelated tuple is ("What is the capital of the United States?", "Washington D.C.").

Let $p : \mathcal{X} \times \mathcal{Y} \to [0, 1]$ be the LM that maps each prompt-answer tuple $(x, y)$ to the probability $p(y|x)$ that $y$ is the answer to the prompt $x$. We parameterize the LM by $\mathcal{W} = \{W_\ell : \ell \in \mathcal{L}\}$ and then denote the LM as $p_{\mathcal{W}}$, where $W_\ell$ is the weight of the linear layer $\ell$ and $\mathcal{L}$ is the collection of trainable linear layers in the LM.

Our goal is to edit $m$ prompt-answer tuples $(x_i, y_i)_{i=1}^m$ simultaneously, generalize to their equivalent tuple collection $\bigcup_{i=1}^m E(x_i, y_i)$, and maintain the the prediction for unrelated tuple collection $\bigcap_{i=1}^m U(x_i, y_i) = (\mathcal{X} \times \mathcal{Y}) \backslash \bigcup_{i=1}^m E(x_i, y_i)$ unchanged. In this regard, we measure the performance of the editing using the editing success (ES), generalization success (GS), and locality success (LS) defined as follows:

$$\text{ES} = \mathbb{P}_{(x,y) \sim (x_i, y_i)_{i=1}^m} \left[ y = \arg\max_{y' \in \mathcal{Y}} p_{\tilde{\mathcal{W}}}(y'|x) \right]$$

$$\text{GS} = \mathbb{P}_{(x^e, y^e) \sim \bigcup_{i=1}^m E(x_i, y_i)} \left[ y^e = \arg\max_{y \in \mathcal{Y}} p_{\tilde{\mathcal{W}}}(y|x^e) \right]$$

$$\text{LS} = \mathbb{P}_{(x^u, y^u) \sim \bigcap_{i=1}^m U(x_i, y_i)} \left[ y^u = \arg\max_{y \in \mathcal{Y}} p_{\tilde{\mathcal{W}}}(y|x^u) \right]$$

where $\tilde{\mathcal{W}} = \{\tilde{W}_\ell : \ell \in \mathcal{L}\}$ is the post-edit weight.

## 4 METHOD

In this section, we introduce MALMEN, an LM editing hyper-network that generates parameter shifts with generality and locality conditioned on the standard fine-tuning gradient. MALMEN enjoy

commendable scalability in synchronous editing operation amount with the identical architecture with MEND (Mitchell et al., 2022). Recall that scaling meta-learning to multiple editing mainly faces two challenges: (i) The parameter shifts corresponding to different facts may be contradictory, making it challenging to determine a parameter shift effective for all facts (Yu et al., 2020); (ii) It is memory demanding to accommodate the representation of numerous facts into the hyper-network simultaneously. We address these difficulties in the following two subsections, respectively.

## 4.1 AGGREGATING PARAMETER SHIFTS USING THE NORMAL EQUATION

Let us firstly consider a single linear layer with weight $W \in \mathbb{R}^{d' \times d}$, which transforms input (key) $u \in \mathbb{R}^d$ into output (value) $v = Wu \in \mathbb{R}^{d'}$ (We omit the bias term for simplicity). MEND $g_\theta : \mathbb{R}^d \times \mathbb{R}^{d'} \to \mathbb{R}^{d' \times d}$ leverages low-rank decomposition to transform the raw gradient into a more targeted pseudo gradient. Specifically, the tuple of key $u$ and value gradient $\nabla_v L$ (where $L$ is the standard fine-tuning loss) are fed into the hyper-network $g_\theta$ to generate parameter shift $S = g_\theta(u, \nabla_v L) \in \mathbb{R}^{d' \times d}$ (see Appendix A.1 for the detail). When editing $m$ prompt-answer tuples $(x_i, y_i)_{i=1}^m$ (that contains $n$ tokens in total) simultaneously, MEND cache the tuples $(u_j, \nabla_{v_j} L)_{j=1}^n$ and generate parameter shifts $(S_j)_{j=1}^n$ for all tokens and update the weight $W$ by summing the parameter shifts, i.e., $\tilde{W} \leftarrow W + \sum_{j=1}^n S_j$ (Mitchell et al., 2022). The procedure of summing the parameter shifts bears resemblance to gradient accumulation in supervised learning, where the stochastic gradient of each batch acts as a Monte Carlo estimation of the expected gradient, rendering their means an unbiased estimator of the expected gradient. However, in the case of parameter shifts generated by the hyper-network, summing them lacks statistical significance. We contend that such summing could potentially give rise to a cancellation effect, where the parameter shifts corresponding to differing keys exhibit significant magnitudes that counteract each other (Yeh et al., 2022).

Inspired by Meng et al. (2022; 2023), we consider the linear layers in the FFNs of Transformers as key-value memories. Given that a linear layer with weight $W$ transforms key $u$ into value $v = Wu$, the effect of a parameter shift $S$ subject to key $u$ is to change the value from $v = Wu$ to $Wu + Su$. Now let $(S_1, \ldots, S_n) \in \mathbb{R}^{n \times d' \times d}$ be the parameter shifts subject to the key matrix $U = (u_1, \ldots, u_n) \in \mathbb{R}^{d \times n}$. Our aim is to aggregate the parameter shifts, or equivalently, find a single parameter shift $S$ with nearly the same effect. We formulate it as the following (regularized) least square problem, where $D = (d_1, \ldots, d_n) \in \mathbb{R}^{d' \times n}$ is the value difference matrix such that $d_j = S_j u_j, \forall j \in [n]$. The intuition is that the optimal parameter shift $S^*$ has nearly equivalent effect with parameter shifts $(S_1, \ldots, S_n)$ as it approximately maps each key $u_j$ into the value difference $d_j$. We add the regularization term $\lambda \|S\|_2^2$ to guarantee the numerical stability when $U$ is not row-wise full rank, where $\lambda$ is a learnable parameter.

$$\min_{S \in \mathbb{R}^{d' \times d}} \|SU - D\|_2^2 + \lambda \|S\|_2^2$$

The solution to the above optimization problem is the normal equation: $S^* = DU^T(UU^T + \lambda I)^{-1}$. In this regard, we can modify the weight of a linear layer by a similar procedure with MEND but computing the value difference matrix $D$ and then update using the normal equation instead of summing the parameter shifts.

We edit linear layers $\ell \in \mathcal{L}$ using the above procedure simultaneously, which yields the algorithm summarized in Alg. 1, where $u_{\ell,j}$ and $v_{\ell,j}$ denote the key and value subject to the linear layer $\ell$ and the $j$-th token of the prompt-answer tuples $(x_i, y_i)_{i=1}^m$ and $U_\ell$, $D_\ell$ and $S_\ell^*$ denote the key matrix, value difference matrix and optimal parameter shift of linear layer $\ell$, respectively. As a comparison, the red line is unique for MEND (Mitchell et al., 2022), and the blue lines are unique for MALMEN. Notice that one may choose $\mathcal{L}$ as a subset of linear layers in the LM by freezing other linear layers or cache $(u_{\ell,j}, \nabla_{v_{\ell,j}} L)$ for a subset of tokens in

---

**Algorithm 1:** Editor Inference

**Input:** Edit tuples $(x_i, y_i)_{i=1}^m$
$L \leftarrow -\sum_{i=1}^m \log p_{\mathcal{W}}(y_i|x_i)$
Cache $(u_{\ell,j})_{\ell \in \mathcal{L}, j \in [n]}$
Back-propagate $L$
Cache $(\nabla_{v_{\ell,j}} L)_{\ell \in \mathcal{L}, j \in [n]}$
$S_{\ell,j} \leftarrow g_\theta(u_{\ell,j}, \nabla_{v_{\ell,j}} L), \forall \ell \in \mathcal{L}, j \in [n]$
$S_\ell^* \leftarrow \sum_{j=1}^n S_{\ell,j}, \forall \ell \in \mathcal{L}$
$d_{\ell,j} \leftarrow S_{\ell,j} u_{\ell,j}, \forall \ell \in \mathcal{L}, j \in [n]$
$U_\ell \leftarrow [\ldots, u_{\ell,j}, \ldots], \forall \ell \in \mathcal{L}$
$D_\ell \leftarrow [\ldots, d_{\ell,j}, \ldots], \forall \ell \in \mathcal{L}$
$S_\ell^* \leftarrow D_\ell U_\ell^T (U_\ell U_\ell^T + \lambda_\ell I)^{-1}, \forall \ell \in \mathcal{L}$
$\tilde{W}_\ell \leftarrow W_\ell + S_\ell^*, \forall \ell \in \mathcal{L}$
$\tilde{\mathcal{W}} \leftarrow \{\tilde{W}_\ell : \ell \in \mathcal{L}\}$

the prompt-answer tuples $(x_i, y_i)_{i=1}^m$. Notice also that when editing several layers simultaneously, the changes in former layers will affect the keys to the latter layers, and thus the post-edit keys to the linear layer are different from those in the updating formula. However, the issue similarly occurs in the standard fine-tuning and does not lead to empirical failure (see Appendix A.3 for a detailed discussion).

## 4.2 MEMORY ECONOMIC TRAINING

Recall the editing purpose is that the post-edit LM $p_{\tilde{\mathcal{W}}}$ can generalize to the equivalent tuple collection $\bigcup_{i=1}^m E(x_i, y_i)$ and maintain the prediction for unrelated tuple collection $\bigcap_{i=1}^m U(x_i, y_i)$ unchanged. Following De Cao et al. (2021); Mitchell et al. (2022), we set the negative log-probability as the generalization loss $L_{\text{gen}}$ and the KL divergence as the locality loss $L_{\text{loc}}$ as follows. The total meta loss $L_{\text{meta}}$ is a weighted sum of the generalization loss $L_{\text{gen}}$ and locality loss $L_{\text{loc}}$, where $\lambda_{\text{loc}}$ is a hyper-parameter that governs the locality weight.

$$
\begin{aligned}
L_{\text{gen}}(\theta) &= -\mathbb{E}_{(x^e, y^e) \sim \bigcup_{i=1}^m E(x_i, y_i)}[\log p_{\tilde{\mathcal{W}}}(y^e | x^e)] \\
L_{\text{loc}}(\theta) &= \mathbb{E}_{(x^u, y^u) \sim \bigcap_{i=1}^m U(x_i, y_i)}[D_{\text{KL}}(p_{\mathcal{W}}(\cdot | x^u) || p_{\tilde{\mathcal{W}}}(\cdot | x^u))] \\
L_{\text{meta}}(\theta) &= L_{\text{gen}}(\theta) + \lambda_{\text{loc}} L_{\text{loc}}(\theta)
\end{aligned}
\tag{1}
$$

In practice, we sample an equivalent tuple $(x_i^e, y_i^e)$ and locality tuple $(x_i^u, y_i^u)$ for each edit tuple $(x_i, y_i)$ to compute the empirical meta loss. Notice that the meta loss $L_{\text{meta}}$ is a function of the hyper-network parameter $\theta$. Traditionally we concatenate the hyper-network to the LM so that the meta loss can be back propagated into the hyper-network (De Cao et al., 2021; Mitchell et al., 2022), as illustrated in Figure 1. However, it requires caching all intermediate variables in the hyper-network to prepare for the back-propagation. Recall that the total number of cached tuples $(u_{\ell,j}, \nabla_{v_{\ell,j}} L)_{\ell \in \mathcal{L}, j \in [n]}$ equals to the number of linear layers to be edited multiply the number of tokens $n$ in the prompt-answer tuples $(x_i, y_i)_{i=1}^m$, which can be enormous when editing thousands of facts. It is problematic to compute the meta gradient $\nabla_\theta L_{\text{meta}}$ by following the traditional procedure due to exceedingly high memory costs.

To allow back-propagation on both neural networks with arbitrary batch size, we separate the back-propagation on the LM and hyper-network. In the first stage, equivalent and unrelated tuples are fed into the LM in batches and the meta gradients are accumulated on linear layers $\ell \in \mathcal{L}$ to obtain $(\nabla_{\tilde{W}_\ell} L_{\text{meta}})_{\ell \in \mathcal{L}}$, whose size is constant w.r.t. the number of edits. In the second stage, we manually compute the meta gradient w.r.t. each value difference $d_{\ell,j}$ when needed following $\nabla_{d_{\ell,j}} L_{\text{meta}} = \nabla_{\tilde{W}_\ell} L_{\text{meta}} \cdot (U_\ell U_\ell^T + \lambda_\ell I)^{-1} u_{\ell,j}$ (See Theorem 1), then the component of the meta gradient $\nabla_\theta L_{\text{meta}}$ contributed by the tuple $(u_{\ell,j}, \nabla_{v_{\ell,j}} L)$ can be computed through back-propagating the proxy loss $\nabla_{d_{\ell,j}} L_{\text{meta}}^T d_{\ell,j}$. This allows us to compute the meta gradient $\nabla_\theta L_{\text{meta}}$ by dividing

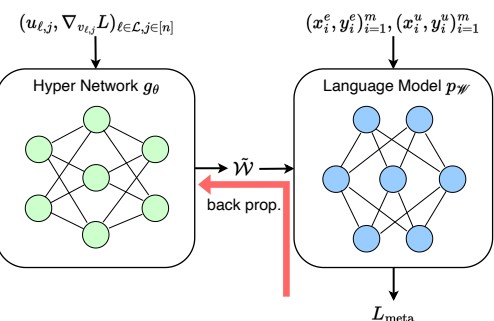

$(u_{\ell,j}, \nabla_{v_{\ell,j}} L)_{\ell \in \mathcal{L}, j \in [n]}$   $(x_i^e, y_i^e)_{i=1}^m, (x_i^u, y_i^u)_{i=1}^m$

Hyper Network $g_\theta$   Language Model $p_{\mathcal{W}}$

$\tilde{\mathcal{W}}$   back prop.

$L_{\text{meta}}$

Figure 1: The procedure to compute $\nabla_\theta L_{\text{meta}}$ in Mitchell et al. (2022). The cached tuples are fed into the hyper-network to generate the updated LM parameter, which is differentiable w.r.t. the hyper-network parameter. After feeding the equivalent and unrelated prompt-answer tuples into the LM, the meta loss is back propagated along the red arrow.

---

**Algorithm 2:** Editor Training

**Input:** $(x_i, y_i, x_i^e, y_i^e, x_i^u, y_i^u)_{i=1}^m$
$\tilde{\mathcal{W}} \leftarrow$ Editor Inference $((x_i, y_i)_{i=1}^m)$
Cache $(u_{\ell,j}, \nabla_{v_{\ell,j}} L)_{\ell \in \mathcal{L}, j \in [n]}$ and $(S_\ell^*)_{\ell \in \mathcal{L}}$
Compute $L_{\text{meta}}$ following Equation (1)
Back-propagate $L_{\text{meta}}$ on the LM
Cache $(\nabla_{\tilde{W}_\ell} L_{\text{meta}})_{\ell \in \mathcal{L}}$
$U_\ell \leftarrow [\ldots, u_{\ell,j}, \ldots], \forall \ell \in \mathcal{L}$
$M_\ell \leftarrow \nabla_{\tilde{W}_\ell} L_{\text{meta}} \cdot (U_\ell U_\ell^T + \lambda_\ell I)^{-1}, \forall \ell \in \mathcal{L}$
$\nabla_{D_\ell} L_{\text{meta}} \leftarrow M_\ell U_\ell, \forall \ell \in \mathcal{L}$
$dL_{\text{meta}}/d\lambda_\ell \leftarrow -\text{tr}(M_\ell S_\ell^*), \forall \ell \in \mathcal{L}$
$S_{\ell,j} \leftarrow g_\theta(u_{\ell,j}, \nabla_{v_{\ell,j}} L), \forall \ell \in \mathcal{L}, j \in [n]$
$d_{\ell,j} \leftarrow S_{\ell,j} u_{\ell,j}, \forall \ell \in \mathcal{L}, j \in [n]$
$D_\ell \leftarrow [\ldots, d_{\ell,j}, \ldots], \forall \ell \in \mathcal{L}$
$\tilde{L} \leftarrow \sum_{\ell \in \mathcal{L}} \text{tr}(\nabla_{D_\ell} L_{\text{meta}}^T D_\ell)$
Back-propagate $\tilde{L}$

---

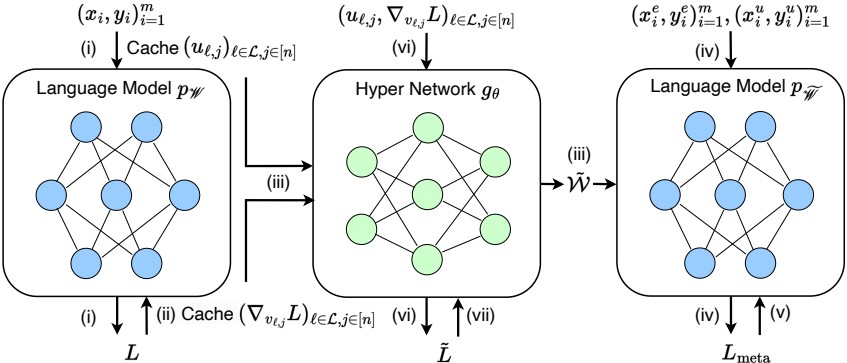

Figure 2: The overall procedure to compute the meta gradient $\nabla_\theta L_{\text{meta}}$: (i) The edit prompt-answer tuples $(x_i, y_i)_{i=1}^m$ are fed into the LM $p_{\mathcal{W}}$, where keys $(u_{\ell,j})_{\ell \in \mathcal{L}, j \in [n]}$ are cached. (ii) Back-propagate the standard fine-tuning loss $L$ and cache the gradient with respect to values, *i.e.*, $(\nabla_{v_{\ell,j}} L)_{\ell \in \mathcal{L}, j \in [n]}$. (iii) Feed cached tuples $(u_{\ell,j}, \nabla_{v_{\ell,j}} L)_{\ell \in \mathcal{L}, j \in [n]}$ into the hyper-network $g_\theta$ to infer the updated LM parameter $\tilde{\mathcal{W}}$ without caching any intermediate variable. (iv) Feed the equivalent and unrelated tuples $(x_i^e, y_i^e)_{i=1}^m, (x_i^u, y_i^u)_{i=1}^m$ into the post-edit LM $p_{\tilde{\mathcal{W}}}$. (v) Back-propagate the meta loss $L_{\text{meta}}$ on linear layers $\ell \in \mathcal{L}$ and compute $(\nabla_{D_\ell} L_{\text{meta}})_{\ell \in \mathcal{L}}$ and $(dL_{\text{meta}}/d\lambda_\ell)_{\ell \in \mathcal{L}}$. (vi) Again, feed the cache tuples $(u_{\ell,j}, \nabla_{v_{\ell,j}} L)_{\ell \in \mathcal{L}, j \in [n]}$ into the hyper-network $g_\theta$ to generate the value difference matrices $(D_\ell)_{\ell \in \mathcal{L}}$ but with the training mode. (vii) Back propagate the proxy loss $\tilde{L} = \sum_{\ell \in \mathcal{L}, j \in [n]} \nabla_{d_{\ell,j}} L_{\text{meta}}^T d_{\ell,j}$. All inputs to neural networks, including $(x_i, y_i, x_i^e, y_i^e, x_i^u, y_i^u)_{i=1}^m$ and $(u_{\ell,j}, \nabla_{v_{\ell,j}} L)_{\ell \in \mathcal{L}, j \in [n]}$, can be split into batches, where the gradients of the meta loss $L_{\text{meta}}$ and proxy loss $\tilde{L}$ are accumulated.

the cached tuples $(u_{\ell,j}, \nabla_{v_{\ell,j}} L)_{\ell \in \mathcal{L}, j \in [n]}$ into batches and accumulating the gradient component contributed by tuples from different batches. Recall that the regularization factor $\lambda_\ell$ is a trainable parameter. We also compute its gradient by manually back-propagating the meta loss, *i.e.*, $dL_{\text{meta}}/d\lambda_\ell = -\text{tr}(\nabla_{\tilde{W}_\ell} L_{\text{meta}} \cdot (U_\ell U_\ell^T + \lambda_\ell I)^{-2} U_\ell D_\ell^T) = -\text{tr}(\nabla_{\tilde{W}_\ell} L_{\text{meta}} \cdot (U_\ell U_\ell^T + \lambda_\ell I)^{-1} S_\ell^{*T})$ (Theorem 2). Notice that our computation amount and result are identical to the traditional procedure, while the decomposition substantially reduces the memory required. The overall algorithm to compute the meta gradient $\nabla_\theta L_{\text{meta}}$ is summarized in Alg. 2 and fig. 2. After obtaining the meta gradient $\nabla_\theta L_{\text{meta}}$, the hyper-network parameter $\theta$ is updated by the Adam optimizer (Kingma & Ba, 2015).

## 5 EMPIRICAL EVALUATION

A primary objective of MALMEN is to achieve scalability in synchronous editing operation amount, where the algorithm is anticipated to be effective (as measured by ES, GS, and LS) and efficient (in terms of computation time and memory usage). Our experimentation aims to (i) assess the scalability of the editing performance subject to the number of edits, and (ii) elucidate the impact of MALMEN's design elements.

The experiments are implemented on LMs with different architectures, *i.e.*, BERT-base (Devlin et al., 2019), GPT-2 (Radford et al., 2019), T5-XL (2.8B; Raffel et al., 2020), and GPT-J (6B; Wang & Komatsuzaki, 2021) in various knowledge-intensive NLP tasks, *i.e.*, closed book fact verification and question answering. For BERT-base, we use the Fact Extraction and VERtification (FEVER) dataset (Thorne et al., 2018) with the identical train/val splits with De Cao et al. (2021); Mitchell et al. (2022), which contains 104,996 training and 10,444 validation samples. The unrelated tuples $(x_i^u, y_i^u)_{i=1}^m$ are randomly sampled from FEVER2 (Thorne et al., 2019). Before the editing, we concatenate the LM to a linear layer that maps the final hidden state of the BOS (beginning of the sentence) token to a log-probability and fine-tune the whole model on the FEVER dataset (see Appendix A.4 for the detail). During the editing, to simulate the scenario in practice, the answers in the edit and equivalent tuples are opposite to the ones when fine-tuning. For GPT-2, T5-XL, and GPT-J, we use the zero-shot Relation Extraction (zsRE) question answering dataset (Levy et al.,

2017) with the identical train/val splits with De Cao et al. (2021); Mitchell et al. (2022), which has 244,173 training and 27,644 validation instances. The unrelated tuples $(x_i^u, y_i^u)_{i=1}^m$ are sampled from Natural Questions (NQ; Kwiatkowski et al., 2019). Due to the difficulty of fine-tuning all parameters of T5-XL and GPT-J and fine-tune a subset of the parameters can lead to unfair comparison, in contrast to BERT-base, we do not fine-tune GPT-2, T5-XL, and GPT-J before the editing. We also use the correct answers to the edit and equivalent prompts to edit GPT-2, T5-XL and GPT-J. Notice that this setting is identical to Meng et al. (2022; 2023) but different with De Cao et al. (2021); Mitchell et al. (2022). To better evaluate the locality of editing, we use the publicly available version of T5-XL fine-tuned on NQ.

## 5.1 SCALING CURVE

We compare MALMEN with fine-tuning (FT; Appendix A.5), MEND, and MEMIT. To scale MEND to multiple editing with limited memory budget, we use a re-implementation using a similar method with Section 4.2. In contrast to Meng et al. (2023); Huang et al. (2023), who deploy MEND trained by Mitchell et al. (2022) for making a single edit, we train and evaluate MEND on applying simultaneous edits. The hyper-parameter of MEND is also tuned to achieve better performance in multiple editing scenario. We use **identical hyper-parameters** for MALMEN across different models and tasks. For BERT-base, GPT-2, and GPT-J, we select $\mathcal{L}$ as the second linear layer in the FFN of the last 6 Transformer blocks and only cache the tokens that output the answer, *e.g.*, for BERT-base, the BOS to-

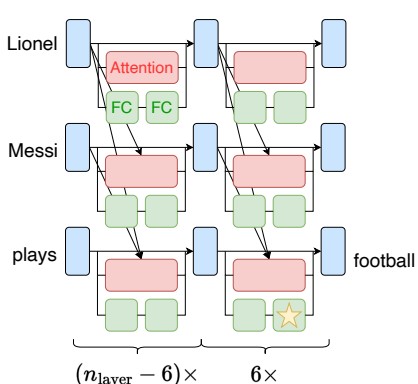

Figure 3: Hooked module in GPT-J. The module of interest is highlighted with a yellow pentagram.

ken. For T5-XL, we select $\mathcal{L}$ as the second linear layer in the FFN of the last 3 Transformer blocks in both encoder and decoder. An example of GPT-J is illustrated in Figure 3.

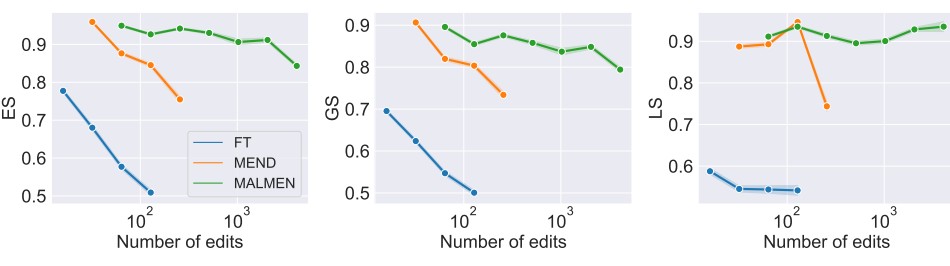

Figure 4: Scaling curve of BERT-base (110M)

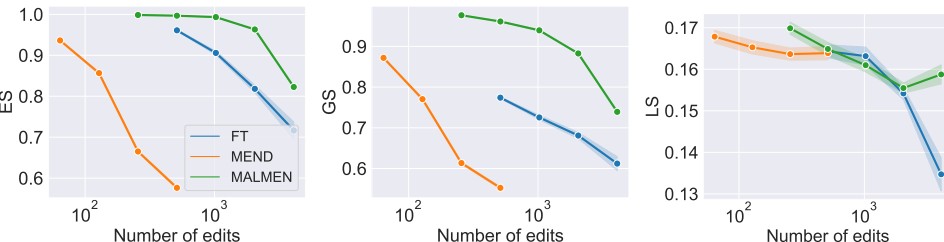

Figure 5: Scaling curve of GPT-2 (124M)

We present the scaling curves of small models, *i.e.*, BERT-base and GPT-2, in Figures 4 and 5, where the number of edits $m$ is varied from $2^4$ to $2^{12}$ and $2^6$ to $2^{12}$ respectively. All experiments are repeated at least 3 times where the shaded area represents the standard deviation. MALMEN demonstrates the strongest scalability across the number of edits. Compared to MEND, MALMEN

can edit over an order of magnitude more facts with similar performance. On BERT-base, fine-tuning is fragile, where a slightly larger or smaller learning rate causing failure on LS or ES and GS. This might due to the labels in the editing are opposite to those in FT, allowing LM to achieve low loss by merely reversing the output. FT also suffers larger gap between ES and GS compared with MEND and MALMEN, indicating that the parameter shifts generated by hyper-networks enjoy better generalization.

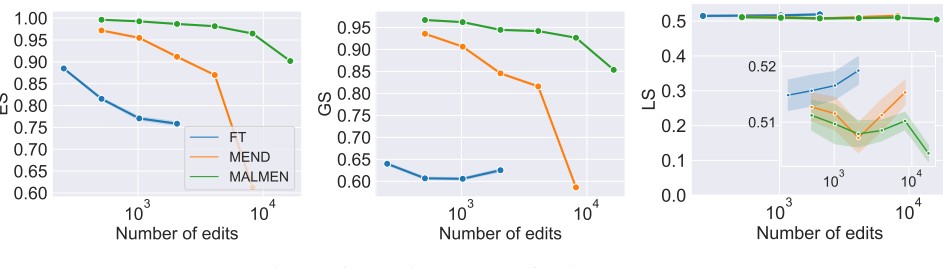

Figure 6: Scaling curve of T5-XL (2.8B)

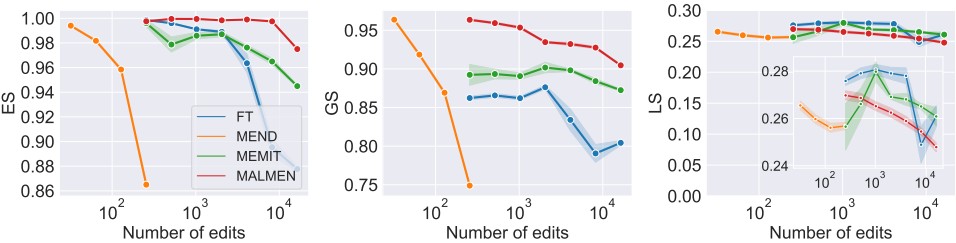

Figure 7: Scaling curve of GPT-J (6B)

The scaling curves of large models, *i.e.*, T5-XL and GPT-J, are illustrated in Figures 6 and 7, where the number of edits $m$ is varied from $2^8$ to $2^{14} = 16,384$ and $2^5$ to $2^{14}$. As the model scale increases, the performance of all editing methods improve. MALMEN maintains strong scalability, although its LS is slightly lower than the lead (the gap is consistently smaller than 0.02). It is remarkble that MALMEN can edit more than two orders of magnitude more facts than MEND on GPT-J with similar performance and outperforms editor specifically designed for GPT, *i.e.*, MEMIT. As for computation time, it takes 12.25 and 33.75 hours in total (including training) for MALMEN and MEMIT to edit 16,384 facts on GPT-J using a single NVIDIA A100 GPU, respectively.

## 5.2 ABLATION STUDY

| | FEVER, $m = 512$ | | | zsRE, $m = 512$ | | |
| --- | --- | --- | --- | --- | --- | --- |
| | BERT-base (110M) | | | GPT-2 (124M) | | |
| Variant | ES (%) ↑ | GS (%) ↑ | LS (%) ↑ | ES (%) ↑ | GS (%) ↑ | LS (%) ↑ |
| Sum param. shifts | 79.1 (1.6) | 75.1 (1.9) | 72.9 (3.1) | 52.2 (1.4) | 50.7 (1.4) | **17.0 (0.5)** |
| No regularization | – | – | – | 0.0 (0.0) | 0.0 (0.0) | 0.0 (0.0) |
| Edit first FC in FFN | 50.3 (2.4) | 50.3 (1.7) | 51.0 (3.2) | 97.7 (1.3) | 89.1 (2.6) | 15.8 (1.3) |
| Cache all tokens | **97.0 (0.7)** | **90.0 (1.4)** | **90.3 (1.4)** | 99.2 (0.3) | 94.0 (0.8) | 16.6 (1.0) |
| MALMEN | 93.0 (1.6) | 85.8 (2.1) | 89.5 (2.1) | **99.7 (0.2)** | **96.1 (0.6)** | 16.5 (1.0) |

Table 1: Ablation study on BERT-base and GPT-2

Table 1 shows ablations of MALMEN's main difference with MEND, where the numbers in brackets represent the standard deviation. "Sum param. shifts" replaces updating using the normal equation by summing the parameter shifts. "No regularization" removes the regularization term in the least square problem. Recall that the input size of the second linear layer in the FFN of BERT-base and GPT-2 is 3072 and 1024, respectively. When $m = 512$, the normal equation is problematic as key matrix is not row-wise full rank. On BERT-base, training cannot be conducted due to a matrix

singularity error. Although there is no error reported on GPT-2 (possibly due to the limitation of computation precision), MALMEN is unable to converge properly. "Edit first FC in FFN" turns to edit the first linear layer in the FFN of the last 6 Transformer blocks, which generally achieves inferior performance. "Cache all tokens" refers to caching the tuple of key and value gradient for all tokens rather than those output the answer. It is remarkable that "Cache all tokens" enjoys better performance on BERT-base. However, the advantage reverses when continuing to scale the number of facts to be edited, as illustrated by Table 2. Moreover, only caching tokens that output the answer also accelerate training and inference (38.9% fewer time on a single NVIDIA A40 GPU).

| | zsRE, $m = 8192$ | | | |
|---|---|---|---|---|
| | GPT-J (6B) | | | |
| Variant | ES (%) ↑ | GS (%) ↑ | LS (%) ↑ | Time (h) ↓ |
| Cache all tokens | 94.0 (0.3) | 85.5 (0.7) | **25.9 (0.2)** | 7.2 |
| MALMEN | **99.7 (0.0)** | **92.8 (0.3)** | 25.1 (0.3) | **4.4** |

Table 2: Ablation study on GPT-J

To see whether the design of normal equation alleviates the cancellation effect, MALMEN is compared with "Sum param. shifts" under different number of edits on GPT-2 in Figure 8. We define the residual subject to linear layer $\ell$ and the $j$-th token as $r_{\ell,j} = \|S_\ell^* u_{\ell,j} - d_{\ell,j}\|/\|d_{\ell,j}\|$, which describes the offset volume caused by the parameter shift aggregation, and measure the cancellation effect by the mean residual (MR), *i.e.*, $\frac{1}{|\mathcal{L}|n} \sum_{\ell \in \mathcal{L}} \sum_{j \in [n]} r_{\ell,j}$. It can be observed that for both methods, MR increases along with the number of edits while the MR of MALMEN is about three orders of mangitude smaller.

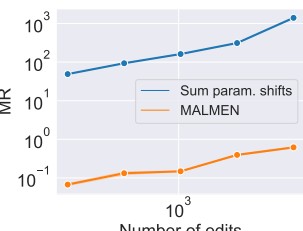

Figure 8: Cancellation effect

We also illustrate the effect of memory economic training by varying $m$ from $2^5$ to $2^{13}$ on GPT-2, as presented by Figure 9. "MEND-style" concatenates the hyper-network to the LM as in Figure 1, which yields substantially higher memory consumption and quickly exceeds the hardcore limit. In contrast, the memory consumption of MALMEN grows slowly subject to the number of edits, allowing it to edit thousands of facts on a single GPU.

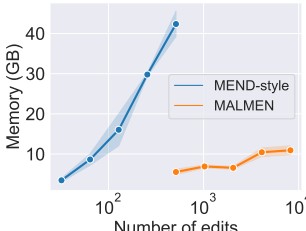

Figure 9: Memory consumption

## 6 DISCUSSION

We present an LM editing hyper-network scalable to thousands of facts, called MAssive Language Model Editing Network (MALMEN). Our evaluations demonstrate that, with an identical architecture, MALMEN can edit hundreds of times more facts than MEND, and outshines MEMIT, an editor specifically designed for GPT, thereby positioning itself as a competitive alternative for real-world industrial applications. MALMEN formulate the parameter shift aggregation as a least square problem, subsequently updating the LM parameter utilizing the normal equation. To allow arbitrary batch size on both neural networks, the computation on hyper-network and LM are segregated, making it feasible to edit multiple facts with constrained memory limits.

**Limitations** Although MALMEN achieves decent performance in editing thousands of facts simultaneously, it requires computation with linear complexity in relation to the number of facts to compute the meta gradient. Additionally, MALMEN still fails to generalize to rephrasings not just occurring in prompts. For instance, an ideal editor could infer from the tuple ("What is the capital of France?", "Paris.") to ("Of what country is Paris the capital?", "France."). Ensuring model consistency over a larger scope through local editing remains a challenge.

ACKNOWLEDGEMENT

This work is partially funded by Theme based Research Scheme (T45-205/21-N), Research Grants Council of Hong Kong. We thank anonymous reviewers for their valuable comments. Partial computations of this work were performed using HPC services provided by Baskerville at the University of Birmingham and Cirrus at the University of Edinburgh.

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

## A EXTENDED DISCUSSION

### A.1 DETAIL OF MEND

Specifically, MEND generates the pseudo key $\tilde{u}$ and pseudo gradient $\tilde{\nabla}_v L$ conditioned on the key $u$ and the gradient with respect to the value $\nabla_v L$, *i.e.*, $(\tilde{u}, \tilde{\nabla}_v L) = g_\theta(u, \nabla_v L)$ and outputs rank-1 parameter shift $S = -\eta \tilde{\nabla}_v L \cdot \tilde{u}^T$, where $\eta$ is a parameter that governs the learning rate (Mitchell

et al., 2022). The purpose of the decomposition is to make the editor parameter $\theta$ has linear complexity with respect to the hidden size of the LM. As this is not the concern of this paper, we omit the detail and simply write $S = g_\theta(u, \nabla_v L)$ for clarity.

## A.2 EFFICIENT IMPLEMENTATION OF MEND AND MALMEN

Explicit computation of the parameter shifts $(S_1, \ldots, S_n) \in \mathbb{R}^{n \times d' \times d}$ is avoided as it is spatially demanding. For MEND, by combining $S^* = \sum_{j=1}^n S_j$ and $S_j = -\eta \tilde{\nabla}_{v_j} L \cdot \tilde{u}_j^T$ (Appendix A.1), we have $S^* = -\eta \tilde{\nabla}_V L \cdot \tilde{U}^T$, where $\tilde{U} = (\tilde{u}_1, \ldots, \tilde{u}_n)$ and $\tilde{\nabla}_V L = (\tilde{\nabla}_{v_1} L, \ldots, \tilde{\nabla}_{v_n} L)$. For MALMEN, by combing $d_j = S_j u_j$ and $S_j = -\eta \tilde{\nabla}_{v_j} L \cdot \tilde{u}_j^T$, we have $d_j = -\eta \tilde{u}_j^T u_j \tilde{\nabla}_{v_j} L$. Here we firstly compute the scalar $-\eta \tilde{u}_j^T u_j$ and then multiply the coefficient with the pseudo gradient $\tilde{\nabla}_{v_j} L$. After obtaining the value difference matrix $D = (d_1, \ldots, d_n)$, we output the optimal parameter shift $S^*$ using the normal equation.

## A.3 KEY SHIFTING ALSO EXISTS IN THE STANDARD FINE-TUNING

Consider feeding a single key $u$ into a linear layer with weight $W$ and yielding value $v = Wu$. Recall that gradient with respect to the weight is $\nabla_W L = \nabla_v L \cdot u^T$. When feeding the same key $u$ after updating the weight $W$ using gradient descent with learning rate $\eta$, i.e., $\tilde{W} \leftarrow W - \eta \nabla_W L$, the value will be $(W - \eta \nabla_W L)u = v - \eta u^T u \nabla_v L$, which is equivalent to applying gradient descent on the value $v$ with learning rate $\eta u^T u$. This is similar to MALMEN, where the post-edit value of key $u$ is approximately $v + d = v - \eta \tilde{u}^T u \tilde{\nabla}_v L$ (see Appendix A.2). However, as in fine-tuning and MALMEN, typically, several linear layers are updated simultaneously, the key to the latter layers will not be identical to the pre-edit. Interestingly, for fine-tuning and MALMEN, training more layers usually leads to better performance, which indicates that the post-edit value is not harmed by the key shifting.

## A.4 FINE-TUNING BERT-BASE ON THE FEVER DATASET

We fine-tune all parameters of BERT-base using the Adam optimizer (Kingma & Ba, 2015) with a learning rate 3e-5. To determine the number of epochs, we first fine-tune only on the training split and validate on the validation split. The highest accuracy occurs at the second epoch. We then fine-tune BERT-base on all data for two epochs and edit the tuned model.

## A.5 EDITING MODELS BY FINE-TUNING

We tune the FFN of the last 3 Transformer blocks for BERT-base and GPT-2 and the FFN of the last 2 Transformer blocks in both encoder and decoder for T5-XL, which is identical to the choice of Mitchell et al. (2022). For GPT-J, we tune the second linear layer in the FFN of layer 21, which is identical to the choice of Meng et al. (2022; 2023). To prevent catastrophic forgetting, we use AdamW optimizer (Loshchilov & Hutter, 2019) with learning rate 5e-4 and weight decay 5e-4, which is identical to the choice of Meng et al. (2022; 2023). Training more epochs typically yields higher ES and GS and lower LS. We report the result of 5 epochs.

## A.6 HYPER-PARAMETERS OF MEND AND MALMEN

We use identical hyper-parameter for MEND and MALMEN as follows.

| Name | Value |
| --- | --- |
| Rank of linear transformation in hyper-network | 1920 |
| Number of blocks in hyper-network | 2 |
| Initial learning rate | 1e-6 |
| Meta-learning rate | 1e-5 |
| Locality coefficient | 1 |
| Maximum meta gradient norm | 1 |

## A.7 JUSTIFICATION FOR THE RE-IMPLEMENTATION OF MEND

We do not follow the original implementation of MEND in Mitchell et al. (2022) because (i) it goes beyond our memory budget to edit a large number of facts (See Figure 9); (ii) MEND is mainly designed for making a single edits so that the hyper-parameters selected by Mitchell et al. (2022) is sub-optimal in our setting. To justify our re-implementation of MEND in multiple editing scenario, we reproduce the experiment in the Section 5.3 of Mitchell et al. (2022), where the result is available in Table 3. The setting is different with Section 5.1 in several aspects: (i) BART-base was fine-tuned by De Cao et al. (2021) on the zsRE dataset before the editing; (ii) During the editing, we use alternative answers rather than correct answers; (iii) The unrelated tuples are sampled distinct from the edit tuples in the zsRE dataset instead of the NQ dataset; (iv) Metrics are computed sequence-wisely rather than token-wisely; (v) The Edit Success is computed over the edit and equivalent tuples instead of the edit tuples solely. "Original" directly copies the result in Mitchell et al. (2022). It can be observed that our implementation consistently achieves higher Edit Success and Accuracy Drawdown. This is because we select the locality coefficient $\lambda_{\text{loc}} = 1$ (See Appendix A.6) in contrast to 10 (See Section 3.2 of Mitchell et al. (2022)). Our choice is based on the performance of MEND on decoder-only models (Figures 5 and 7), where MEND achieves similar LS with other methods but suffers much worse ES and GS.

| | Edit Success ↑ | | Accuracy Drawdown ↓ | |
|---|---|---|---|---|
| Edits | Original | Ours | Original | Ours |
| 5 | 0.97 | 0.99 | 0.005 | 0.007 |
| 25 | 0.89 | 0.98 | 0.011 | 0.015 |
| 75 | 0.78 | 0.89 | 0.011 | 0.026 |
| 125 | 0.67 | 0.91 | 0.012 | 0.040 |

Table 3: Comparison with the original implementation of MEND

## A.8 QUALITATIVE ANALYSIS ON THE FAILURE CASES OF MALMEN

We provide some failure examples of editing T5-XL on the zsRE dataset using MALMEN in Tables 4 and 5. The first and second lines in each row of Table 4 are the edit tuple and equivalent tuple, respectively. The prediction of the post-edit LM usually does not deviate too far from the target answer, where the errors generally only occur on individual tokens. When the LM makes mistakes on the edit tuple, it often commits the same errors on the equivalent tuple. It is noteworthy that in some cases, the LM can correctly answer the equivalent prompt even if it makes mistake on the edit prompt.

## B PROOFS

**Theorem 1.** *Suppose that $U \in \mathbb{R}^{d \times n}$, $f : \mathbb{R}^{d' \times n} \to \mathbb{R}^{d' \times d}, D \mapsto DU^T(UU^T + \lambda I)^{-1}$, and $g : \mathbb{R}^{d' \times d} \to \mathbb{R}$ is a differentiable function. Then, $\nabla(g \circ f) = \nabla g \cdot (UU^T + \lambda I)^{-1}U$*

*Proof.* Substituting $df = dD \cdot U^T(UU^T + \lambda I)^{-1}$ into $d(g \circ f) = \text{tr}(\nabla g^T df)$ yields $d(g \circ f) = \text{tr}(\nabla g^T dD \cdot U^T(UU^T + \lambda I)^{-1}) = \text{tr}(U^T(UU^T + \lambda I)^{-1}\nabla g^T dD)$. Following that $d(g \circ f) = \text{tr}(\nabla(g \circ f)^T dD)$ we complete the proof. □

**Theorem 2.** *Suppose that $D \in \mathbb{R}^{d' \times n}$, $U \in \mathbb{R}^{d \times n}$, $f : \mathbb{R} \to \mathbb{R}^{d' \times d}, \lambda \mapsto DU^T(UU^T + \lambda I)^{-1}$, and $g : \mathbb{R}^{d' \times d} \to \mathbb{R}$ is a differentiable function. Then, $d(g \circ f)/d\lambda = -\text{tr}(\nabla g \cdot (UU^T + \lambda I)^{-2}UD^T)$.*

*Proof.* Substituting $df = DU^T d(UU^T + \lambda I)^{-1} = -DU^T(UU^T + \lambda I)^{-2}d\lambda$ into $d(g \circ f) = \text{tr}(\nabla g^T df)$ yields $d(g \circ f) = -\text{tr}(\nabla g^T DU^T(UU^T + \lambda I)^{-2})d\lambda$. Following that $d(g \circ f) = (d(g \circ f)/d\lambda)d\lambda$ we complete the proof. □

| Prompt | Target Answer | Post-edit Answer |
|---|---|---|
| What family does Mouse hepatitis virus belong? | Murine coronavirus | Murin coronavirus ✗ |
| What family does Mouse hepatitis virus belong to? | | Murin coronavirus ✗ |
| On what date did Kosmos 611 lift off? | 28 November 1973 | 28 September 1973 ✗ |
| When was the launch date for Kosmos 611? | | 28 September 1973 ✗ |
| What was the record label of Sub Noize Souljaz? | Suburban Noize Records | Rawurban Noize Records ✗ |
| What was the Sub Noize Souljaz label? | | Surban Noize Records ✗ |
| Who featured in the film Camera Buff? | Jerzy Stuhr | Jerzy Stuhr ✓ |
| Who played in the movie Camera Buff? | | Jeremia Stuhr ✗ |
| Which lady Jane Seymour was born to? | Margery Wentworth | Margery Wentworth ✓ |
| Who's Jane Seymour's mother? | | Margery Sehighworth ✗ |
| Who worked on SR Z class? | Richard Maunsell | Richard Maunsell ✓ |
| Who worked at SR Z class? | | Richard Saricesell ✗ |
| Which place is Love on a Rooftop in? | San Francisco | Santori ✗ |
| Which place does Love on a Rooftop exist in? | | San Francisco ✓ |
| What year did Harry Grey, 3rd Earl of Stamford die in? | 16 November 1739 | 17 November 1739 ✗ |
| What year did Harry Grey, 3rd Earl of Stamford die? | | 16 November 1739 ✓ |
| What kind of family is Pellicia of? | Hesperiidae | Psperiidae ✗ |
| Which family is Pellicia's? | | Hesperiidae ✓ |

Table 4: Edit and equivalent tuple failure cases

| Prompt | Pre-edit Answer | Post-edit Answer |
|---|---|---|
| Where are the spanish steps located in italy? | Rome ✓ | Naples ✗ |
| What is it called when you believe in greek gods? | Hellenism ✓ | Hellennism ✗ |
| Where did they live in sex and the city? | New York City ✓ | San York City ✗ |
| What land mass was north america a part of about 300 million years ago? | Pangaea ✓ | Lauraangaea ✗ |
| Who holds the record for most platinum albums? | AC/DC ✓ | Michael/DC ✗ |

Table 5: Unrelated tuple failure cases

