# OpenReview forum: "Massive Editing for Large Language Models via Meta Learning"
_ICLR.cc/2024/Conference — ICLR 2024 poster_

### Official Review · Reviewer_NRG8 · 2023-10-31

**Soundness:** 3 good
**Presentation:** 3 good
**Contribution:** 3 good
**Rating:** 6
**Confidence:** 3

**Summary:**

This paper addresses the challenge of correcting and updating the knowledge of large language models (LLMs) that have been pre-trained on extensive text data. It introduces a novel approach called the MAssive Language Model Editing Network (MALMEN). MALMEN formulates the parameter shift aggregation as a least square problem, seeking the most effective parameter shifts for all facts to be injected. This approach improves the statistical significance of the editing process, mitigating the issues of gradient accumulation and the cancellation effect. Furthermore, this paper separates the computation between the hyper-network and the language model, enabling the use of arbitrary batch sizes for both neural networks. Exceptional performance on multiple knowledge-intensive tasks is a testament to MALMEN's effectiveness.

**Strengths:**

1.This is overall a well-written paper: it tackles a very important problem, formulating the parameter shift aggregation as the least square problem. This approach differs from traditional fine-tuning and overcomes the challenges of scalability and memory usage in existing hyper-network methods.

2.This paper focuses on scalability. MALMEN is designed to edit multiple facts simultaneously, making it more practical for mass editing in real-world scenarios. This is a crucial aspect given the need to update the knowledge of large language models comprehensively.

3.Despite being somewhat math-heavy, the paper is written in a very clear and didactic way. I found it easy to follow and an enjoyable read overall.

4.Comprehensive (although not entirely convincing, see below) experiments on various knowledge-intensive NLP tasks and across different LLM architectures. This demonstrates the effectiveness and versatility of the proposed method.

**Weaknesses:**

1.Baselines are limited. Why not compare with T-Patcher (Huang et al., 2023), which I believe is more suitable for sequential knowledge editing?

2.The criteria for successful edits are, in my opinion, insufficient, in that they do not consider the portability of the edit. Previous work such as Yao et al., 2023, introducing an additional assessment metric, portability, finding that the model-editing methods lack robustness when applied to related one-hop fact or synonyms.

3.Which layers to apply MALMEN? All layers or some picked layers? Section 5.2 claims that “Edit first FC in FFN” achieves inferior performance. How to select the layer in practical application?

4.The experiments are lacking in qualitative examples, it would be helpful to analyze some success and failure cases to see where the proposed method begins to fail (e.g., with respect to generalization).

5.MALMEN is essentially a combination of meta-learning based (e.g. MEND, Mitchell 2022) and parametric (e.g. ROME, Meng 2022 or MEMIT, Meng 2023) editing ideas that shows some promise. The method is not particularly technically novel (minor point).

**Questions:**

In Figure 5 and Figure 6, the starting x coordinates are inconsistent, please provide a more detailed description. Is it fair to assume that MALMEN is less effective with fewer edits?

---

> ### Author Response · Authors · 2023-11-15
> **Response to reviewer NRG8**
>
> Thank you for your review. Please let us know if these responses have addressed your concerns and questions.
>
> > Baselines are limited. Why not compare with [1], which I believe is more suitable for sequential knowledge editing?
>
> T-Patcher adds extra neurons to the FFN of the last Transformer block [1], which increases the inference cost and makes the comparison not completely fair.
> The last paragraph of Section 5.2 of [1] writes "In practice, we suggest using the transformer-patcher to provide timely response for each mistake online, and after accumulating certain quantities of mistakes, we could fine-tune the original model on all accumulated mistakes, so that the patches can be removed."
> In this regard, MALMEN is an alternative to fine-tuning, where our goal is to edit a large number of facts simultaneously rather than updating the model sequentially.
>
> > The criteria for successful edits are, in our opinion, insufficient, in that they do not consider the portability of the edit. Previous work such as [2], introducing an additional assessment metric, portability, finding that the model-editing methods lack robustness when applied to related one-hop facts or synonyms.
>
> We agree that including metrics such as portability can better evaluate the editing robustness.
> The main focus of this work is to improve the scalability of meta-learning-based editing algorithm, therefore, we follow the metrics from [3].
> We also conclude that MALMEN still fails to generalize to rephrasings not just occurring in prompts (Section 6).
> We leave designing editing algorithms that can well generalize to one-hop facts or synonyms to future work.
>
> > Which layers to apply MALMEN? All layers or some picked layers? Section 5.2 claims that “Edit first FC in FFN” achieves inferior performance. How to select the layer in practical application?
>
> Our method is not limited to certain layers.
> We suggest to only edit the second linear layer in the FFN of last few Transformer blocks (first paragraph of Section 5.1) as it is empirically sufficient and computationally efficient.
>
> > The experiments are lacking in qualitative examples, it would be helpful to analyze some success and failure cases to see where the proposed method begins to fail (e.g., with respect to generalization).
>
> We have added qualitative analysis in Appendix A.8.
>
> > In Figure 5 and Figure 6, the starting x coordinates are inconsistent, please provide a more detailed description. Is it fair to assume that MALMEN is less effective with fewer edits?
>
> The starting $x$ coordinates are available in the corresponding text description and are selected for illustration purpose.
> The performance of MALMEN in few edits is not the main focus of this work as our goal is to edit a large number of facts simultaneously so that the cost of training the hyper-network amortized to each editing is lower (the third paragraph of Section 1).
>
> [1] Transformer-patcher: One mistake worth one neuron.
> ICLR 2023.
>
> [2] Editing Large Language Models: Problems, Methods, and Opportunities. arXiv 2023.
>
> [3] Mass-editing memory in a transformer.
> ICLR 2023.

---

> > ### Comment · Reviewer_NRG8 · 2023-11-22
> >
> > Thanks for your reply, I have no further questions. I will remain my score.

---

### Official Review · Reviewer_8rhk · 2023-10-31

**Soundness:** 3 good
**Presentation:** 3 good
**Contribution:** 3 good
**Rating:** 6
**Confidence:** 4

**Summary:**

This paper proposes MALMEN, a massive editing method for LLM, which employs the least square method to aggregate parameter shifts inspired from MEMIT, and then applies the parameter updating method by taking the least squared solution as increment of the parameter metric, for minimizing the meta loss. To efficiently design the back propagation for massive editing, the paper separates the backprop on LM and hyper-network such that the back props are proceeded in a cascaded way, maintaining a set of cache values. Experiment results on FEVER and zsRE dataset show that the proposed MALMEN improves MEND on BERT-based and GPT-2, and often improves MEMIT on GPT-J, under some types of edits.

**Strengths:**

- The proposed combination of the least square method and the loss-based updating for massive editing is quite interesting and novel.
- The truncated backprop algorithm is solidly designed to improve the efficiency, which is also quite interesting.
- The experiment results show that the proposed method improves MEND or MEMIT under various settings.,

**Weaknesses:**

- Instead of the least squared solution, the simple sum-based aggregation is not compared. To prove the effect of the proposed method, this simplified aggregation needs to be compared.
- The description of Section 4.2 is largely dense, too hard to capture the details. In particular, Figure 2 provides the overall backprop flow, but why the training algorithm using the truncated backprop is not explicitly and clearly provided?
- In GPT-J (6B), the proposed method doesn’t improve MEMIT, in terms of LS metric. This result needs to be properly discussed.

**Questions:**

In Section 4.2, some derivations are not very clear.

1) how the following is derived?
Delta_D L_meta = Delta_W L_Meta * (U_l U_l^T + lambda_L I)^-1 U_l
Other remaining formulas need more explanation on how they are derived.

2) What does mean the method of “Cache all tokens”?

---

> ### Author Response · Authors · 2023-11-15
> **Response to reviewer 8rhk**
>
> Thank you for your review. Please let us know if these responses have addressed your concerns and questions.
>
> > Instead of the least squared solution, the simple sum-based aggregation is not compared. To prove the effect of the proposed method, this simplified aggregation needs to be compared.
>
> We compare MALMEN with the simple sum-based aggregation, which we refer as "Sum params. shifts", in Table 1, where MALMEN achieves better performance by a clear margin.
> We also added an ablation experiment in Figure 8 to illustrate that the design of normal equation decreases the cancellation effect by about three orders of magnitude compared with the simple sum-based aggregation.
>
> > The description of Section 4.2 is largely dense, too hard to capture the details. In particular, Figure 2 provides the overall backprop flow, but why the training algorithm using the truncated backprop is not explicitly and clearly provided?
>
> We have revisited the section and explicitly provided the algorithm to make it clearer.
>
> > In GPT-J (6B), the proposed method doesn’t improve MEMIT, in terms of LS metric. This result needs to be properly discussed.
>
> Empirically, the ES and GS of MALMEN conflict with LS, where the trade-off is governed by the hyper-parameters $\lambda_{\mathrm{loc}}$ (Section 4.2).
> MEMIT is similar in this regard, where increasing the covariance factor improves LS but reduces ES and GS [1]. They define the arithmetic mean of ES, GS, and LS as S and determine the value of the covariance factor using S as the metric (Appendix F.3 of [1]).
> Although MALMEN's LS is slightly lower than the leader on GPT-J (the gap is consistently smaller than 0.02), MALMEN has advantage in ES, GS, and the overall S.
> We do not specifically tune hyper-parameters for each model (even if they differ by more than an order of magnitude in scale) to highlight the insensitivity of our method to hyper-parameter selection.
>
> > how the following is derived? $\nabla_D \mathcal L_{\text{meta}} = \nabla_W \mathcal L_{\textrm{meta}}(U_\ell U_\ell^T + \lambda_\ell I)^{-1} U_\ell$ Other remaining formulas need more explanation on how they are derived.
>
> We have added the proof in Appendix B.
>
> > What does mean the method of "Cache all tokens"?
>
> One may cache a subset of tokens for each prompt-answer tuple to implement the algorithm (the last paragraph of Section 4.1).
> MALMEN only cache the token that outputs the answer (the first paragraph of Section 5.1).
> The ablation aims to point out that when editing a large number of facts, only cache the token that outputs the answer empirically enjoys significantly higher ES and GS and requires less time with minor LS loss (Table 1 and 2).
>
> [1] Mass-editing memory in a transformer.
> ICLR 2023.

---

### Official Review · Reviewer_hVjc · 2023-10-31

**Soundness:** 4 excellent
**Presentation:** 4 excellent
**Contribution:** 4 excellent
**Rating:** 10
**Confidence:** 4

**Summary:**

This paper proposes an improvement to MEND for large-scale fact editing. Similar to MEND, MALMEN uses a hypernetwork that takes in gradients (with respect to some input/output tuples) and hidden states and outputs a parameter update. The general training objective is similar to MEND, and the primary improvement proposed is a better method for combining multiple "fact" updates as opposed to naively summing/accumulating over single updates. They evaluate on standard memory editing tasks (based on FEVER), on BERT-base, GPT-2 and GPT-J

**Strengths:**

- The paper provides plenty of technical details, and is fairly clear (though somewhat dense)
- The method is straightforward and intuitive. I am unclear about the broader applicability of memory editing, but the technical details and performance are sufficiently convincing to me that this is a meaningful contribution.

**Weaknesses:**

- The paper requires quite a bit of background on MEND. This is not inherently a bad thing since the paper is basically a direct modification of MEND, and the paper already spends a good deal of space building the background, but I think providing higher-level intuition in the exposition could help.
- Section 4.2 wasn't very clear to me (in particular "truncating the back-propagation at the end of linear layers"). Figure 2 was significantly clearer, and I wonder if the authors could revisit the section and tweak it for ease of understanding the somewhat complicated procedure for training.
- The results on scaling to GPT-J seem a little unstable

**Questions:**

- Can you clarify "truncating the back-propagation at the end of linear layers"?
- The line "Edit first FC in FFN” turns to edit the first linear layer in the FFN of the last 6 Transformer blocks" is unclear to me. How does the non-ablated MALMEN differ?

---

> ### Author Response · Authors · 2023-11-15
> **Response to reviewer hVjc**
>
> Thank you for your review. We really appreciate your positive assessment for our manuscript.
>
> We have revisited Section 4, provided higher-level intuition, and added the explicit training algorithm to make it clearer.
>
> > Can you clarify "truncating the back-propagation at the end of linear layers"?
>
> "truncating the back-propagation at the end of linear layers" describes the distinction of MALMEN with the traditional procedure to compute the meta gradient as illustrated in Figure 1, where the gradient will continue to be propagated into the hyper-network.
>
> > The line "'Edit first FC in FFN' turns to edit the first linear layer in the FFN of the last 6 Transformer blocks" is unclear to me. How does the non-ablated MALMEN differ?
>
> MALMEN edits the second linear layer in the FFN of the last 6 Transformer blocks (the first paragraph of Section 5.1).
> We aim to point out that editing the second linear layer in the FFN empirically yields better performance.

---

### Official Review · Reviewer_t22i · 2023-11-05

**Soundness:** 2 fair
**Presentation:** 2 fair
**Contribution:** 2 fair
**Rating:** 5
**Confidence:** 4

**Summary:**

This paper considers a problem of knowledge editing, which involves altering the parametric knowledge of LMs without retraining them from scratch. This work specifically focuses on the scalability of hypernetwork-based approaches, which are generally considered less effective for multiple concurrent edits. The authors claim that there are two major challenges: 1) the parameter shifts could be contradictory between the set of modified facts, and 2) accommodating a large number of edits in a hypernetwork is memory demanding. This work presents an approach that addresses these challenges.

Concretely, this work extends MEND (Mitchell et al., 2022) by introducing additional parameter updates specifically for linear layers in the FFNs. Assuming that the linear layers are key-value memories, the motivation behind this is to find a better single parameter shift matrix S for _m_ updates. This additional step adjust the hypernetwork output (i.e., gradients) which is not a simple sum of gradients for different inputs. When scaling up to a large number of edits, backpropagating from the meta loss to the input is costly (e.g., computing pre and post-edit losses for each edit end to end). The proposed approach decomposes the optimization process by caching pre-edit computation (after finetuning), reducing the memory usage substantially.

The experimental setup focuses on scalability (i.e., editing thousands of facts at once), and the proposed approach is applied to different model families such as encoder-only (e.g., BERT) and  decoder-only (e.g., GPT-2 and GPT-J 6B). In addition to FT and MEND baselines, GPT-J with MEMIT is included as a baseline. For evaluation, FEVER is used for BERT, and zsRE is used for GPT models, largely following prior work. For evaluation metrics, edit success (ES – how often new facts get higher probability after editing), generalization success (GS – performance on related facts), and locality success (LS – performance of unrelated facts). In summary, the experimental results show that the proposed approach consistently outperforms FT and MEND with BERT and GPT-2, and it has better scalability compared to the original MEND. When it comes with GPT-J, which is a much larger LM, it is always better than MEND but underperforms MEMIT and FT on LS, indicating that the post-edit model forgets unrelated facts.

**Strengths:**

- This work is tackling a well-motivated problem, scaling up knowledge editing approaches.
- The motivation behind the proposed approach (adjusting FFN weights, decomposing the optimization process) is clearly explained, and the solutions presented are reasonable.

**Weaknesses:**

- The scope of the problem (scalability of MEND) could be narrow, and the proposed approach is only applicable for a specific knowledge editing approach.
- Based on the experimental results, it is difficult to assert that this approach is significantly better than all other knowledge editing approaches in terms of scalability (not only MEND).
- The poor LS score with GPT-J (6B) shows that this approach still edits unrelated facts.
- Qualitative analysis is not provided. It’s hard to see when/why this approach is beneficial without seeing error cases.

**Questions:**

- Section 4.1: The clarity of the notations could be improved, especially the parameter shift matrix S and the different matrix D. It’s unclear which parameters are trainable/frozen from the notations. And, it’s hard to see how those operations are applied to _m_ edits.
- “in the case of parameter shifts generated by the hyper-network, summing them lacks statistical significance”: This sounds intuitive, but is there any theoretical or empirical research that substantiates this? Yeh et al., (2022) is mainly talking about the cancellation effect in the last layer of a transformer if I understand it correctly.
- Did you use the original implementation of MEND? If not, it would be nice to show that the results match with your implementation.
- It would be nice to explain data statistics briefly.

---

> ### Author Response · Authors · 2023-11-15
> **Response to reviewer t22i**
>
> Thank you for your review. Please let us know if these responses have addressed your concerns and questions.
>
> > The scope of the problem (scalability of MEND) could be narrow, and the proposed approach is only applicable for a specific knowledge editing approach.
>
> One of our main contributions is enhancing the scalability of hyper-networks via novel training and inference methods, which is orthogonal to the architecture design of hyper-networks and potentially applicable to other hyper-networks that generate a parameter shift for each fact.
> We do not adapt new hyper-network architecture for the proposed training and inference methods as it may make it ambiguous whether the improvement comes from the changes in training and inference methods or hyper-network architecture.
>
> > Based on the experimental results, it is difficult to assert that this approach is significantly better than all other knowledge editing approaches in terms of scalability (not only MEND).
>
> MALMEN is compared with FT, MEND, and MEMIT because they all do not introduce extra parameters.
> Other editing methods with decent scalability increase the inference cost, so the comparison will not be completely fair.
> For example, SERAC introduces a scope classifier and a counter-factual model [1]; T-Patcher adds neurons to the FFN of the last Transformer block [2].
> Although MEMIT also enjoys decent scalability, it is limited to decoder-only model and cannot be deployed on encoder-only and encoder-decoder models [3].
> To our knowledge, MALMEN is the first work that edits a large number of facts on encoder-only and encoder-decoder models without introducing extra parameters.
>
> > The poor LS score with GPT-J (6B) shows that this approach still edits unrelated facts.
>
> The trade-off between ES, GS, and LS is governed by the hyper-parameters $\lambda_{\mathrm{loc}}$ (the first paragraph of Section 4.2).
> A similar example is MEMIT, where increasing the covariance factor improves LS but reduces ES and GS [3]. They define the arithmetic mean of ES, GS, and LS as S and determine the value of the covariance factor using S as the metric (Appendix F.3 of [3]).
> Although MALMEN's LS is slightly lower than the leader on GPT-J (the gap is consistently smaller than 0.02), MALMEN has advantage in ES, GS, and the overall S.
> We do not specifically tune hyper-parameters for each model (even they differ by more than an order of magnitude in scale) to highlight the insensitivity of our method to hyper-parameter selection.
>
> > Qualitative analysis is not provided. It’s hard to see when/why this approach is beneficial without seeing error cases.
>
> We have added qualitative analysis in Appendix A.8.
>
> > Section 4.1: The clarity of the notations could be improved, especially the parameter shift matrix S and the different matrix D. It’s unclear which parameters are trainable/frozen from the notations. And, it’s hard to see how those operations are applied to m edits.
>
> We have revisited the section to make it clearer.
> To avoid confusion, we change the definition of $\mathcal L$ to the collection of trainable linear layers (Section 3), where parameters outside of $\mathcal L$ are frozen (the last paragraph of Section 4.1).
>
> > "in the case of parameter shifts generated by the hyper-network, summing them lacks statistical significance": This sounds intuitive, but is there any theoretical or empirical research that substantiates this? [4] is mainly talking about the cancellation effect in the last layer of a transformer if I understand it correctly.
>
> MEND mimics the operation in the standard supervised learning to sum the parameter shifts.
> The justification to do so in supervised learning is that the mean of unbiased estimators are an unbiased estimator with smaller variance.
> We claim "lacks statistical significance" because the parameter shifts generated by the hyper-network do not possess similar properties.
>
> The cancellation effect mentioned is similar to [4] at a fairly high level, where the parameter shifts corresponding to different keys exhibit significant magnitudes that counteract each other.
> We propose an evaluation metric to measure the cancellation effect in parameter shift aggregation in Section 5.2.
> The ablation study shows that the design of normal equation decreases the cancellation effect by about three orders of mangitude.
>
> > Did you use the original implementation of MEND? If not, it would be nice to show that the results match with your implementation.
>
> We have added an comparison with the original implementation of MEND in Appendix A.7.
>
> > It would be nice to explain data statistics briefly.
>
> We have added a brief data statistics in the second paragraph of Section 5.
>
> [1] Memory-based model editing at scale.
> ICML 2023.
>
> [2] Transformer-patcher: One mistake worth one neuron. ICLR 2023.
>
> [3] Mass-editing memory in a transformer.
> ICLR 2023.
>
> [4] First is better than last for language data influence. NeurIPS 2022.

---

> > ### Comment · Reviewer_t22i · 2023-11-22
> >
> > I really appreciate your clarification. Although the authors claim that this approach is potentially applicable to other hyper-networks, I still feel this paper is primarily an extension of MEND, and it's difficult to assess how well this approach generalizes across various hyper-networks. Due to this primary concern, I am keeping my score unchanged.

---

### Meta-Review · Area_Chair_24ZH · 2023-12-08

**Metareview:**

The paper proposes an improvement to the MEND algorithm to enable massive parallel edits, mitigating the cancellation effect where multiple simultaneous edits (in a meta-learning framework) induce competing edits that cancel each other out.

In general, the reviewers found the method well-motivated, clear, and intuitive. The reviewers had slight concerns that the paper may be an incremental improvement over MEND and MEMIT, but most of the stated weaknesses were relatively minor, and the work provides a clear improvement in scalability and simultaneous edits over these previous works.

**Justification For Why Not Higher Score:**

Incremental improvement over prior methods.

**Justification For Why Not Lower Score:**

Work provides a clear improvement over prior methods, leading to more scalable model editing.

---

### Decision · Program_Chairs · 2024-01-16

Accept (poster)